# Social-Ecological Examination of Non-Consensual Sexting Perpetration among U.S. Adolescents

**DOI:** 10.3390/ijerph17249477

**Published:** 2020-12-17

**Authors:** Alberto Valido, Dorothy L. Espelage, Jun Sung Hong, Matthew Rivas-Koehl, Luz E. Robinson

**Affiliations:** 1School of Education, University of North Carolina at Chapel Hill, Chapel Hill, NC 27514, USA; espelage@unc.edu (D.L.E.); luzeli@unc.edu (L.E.R.); 2School of Social Work, Wayne State University, Detroit, MI 48202, USA; fl4684@wayne.edu; 3Department of Human Development and Family Sciences, Texas Tech University, Lubbock, TX 79409, USA; Matthew.Rivas-Koehl@ttu.edu

**Keywords:** sexting, adolescent, social-ecology, predictors

## Abstract

Adolescent sexting is a serious public health concern and is associated with adverse psychosocial outcomes, including depression, anxiety, low self-esteem, declining academic performance, and health problems. Effective prevention of sexting requires a comprehensive and deep understanding of the multiple contexts whereby sexting is likely to occur. The present study explores individual and contextual risk and protective factors that are associated with sexting behavior among a large sample of adolescents. Participants were high school students in midwestern U.S. (*N* = 2501; LGB *n* = 309, 76.4% female; non-LGB *n* = 2192, 47.4% female) who completed self-report measures of sexting and risk (e.g., pornography exposure, impulsivity) and protective (e.g., social support) factors. Path analysis models were conducted with the sexting outcome for groups of LGB and non-LGB students. Among LGB students, results indicated a significant association between sexting and parental monitoring (b = −0.08, *p* < 0.01); pornography exposure (b = 0.13, *p* < 0.05); dating partners (b = 0.01, *p* < *0*.01); bullying perpetration (b = 0.17, *p* < 0.001); and delinquency (b = 0.13; *p* < 0.001). Among non-LGB students, significant associations were found between sexting and alcohol/substance use (b = 0.05, *p* < 0.001); bullying (b = 0.08, *p* < 0.001); and delinquency (b = 0.06, *p* < 0.001). Moderation analyses suggest that parental monitoring may have a buffering effect between sexting and several risk factors. Recommendations for practitioners include considering the protective factors of sexting perpetration and encouraging appropriate levels of parental monitoring and the continued importance of bullying and alcohol and drug prevention programming to decrease risk factors of sexting perpetration.

## 1. Introduction

Teen sexting, which is defined as electronically sending sexually explicit images, has received a significant amount of research attention in recent years [1]. A recent United States study of a national estimate of youth indicated that of the 1560 internet users (ages 10–17), 7.1% of youth had appeared in or created nude or nearly nude photos [2]. According to Patchin and Hinduja’s [3] study, which included a national U.S. sample of 5593 middle and high school students, 13% of the students had reported sending a sexually explicit photo via text message. Not all youth who engage in sexting report negative associations [4] and it has been argued that sexting may even be a normative developmental behavior as adolescents seek intimate connections [5]. Even so, an overwhelming amount of literature documents continued concern regarding youth’s involvement in sexting because of the negative outcomes that are frequently found to be associated with such behaviors, as well as the legal ramifications that youth may face as a result. As such, sexting is a serious public health concern, as an emerging body of research has demonstrated that sexting is associated with adverse psychosocial outcomes, including depression, anxiety, low self-esteem, declining academic performance, and health problems [1,6,7,8,9]. In addition, there are considerable legal consequences that could result from sexting, as legal opinions on the matter are contentious; a number of legal cases regarding sexting have resulted in adolescents being charged with violations of child pornography laws. Legal scholars emphasize the importance of furthering knowledge on this phenomenon to inform future policymaking and adjudication [10]. Effectively preventing and intervening in sexting is of importance to practitioners working with adolescents and requires a comprehensive and deep understanding of the multiple contexts whereby sexting is likely to occur. The present study explores individual and contextual risk and protective factors that are associated with sexting behavior among a large sample of adolescents. Importantly, scholars have noted the problematic conflation of consensual and non-consensual sexting in the research literature [11], and the current study seeks to advance sexting literature by explicitly focusing on the effects of non-consensual sexting.

### 1.1. Social-Ecological Systems Framework

Bronfenbrenner’s [12] seminal ecological systems framework has been proposed as the preferred framework for examining the determinants of multiple forms of youth violence and adolescent risky behaviors. He proposed that human development and behavior are a product of the interactive interplay between the individual and multiple layers of one’s social environment. This framework postulates that risky behaviors, like sexting, result from risk factors situated in multiple systems or domains (microsystem, mesosystem, exosystem, and macrosystem; [12]). Further, engagement in risky behaviors is also attenuated when there are significant protective or promotive factors at play in each of these domains. This ecology framework guided the current study of predictors of adolescent sexting.

### 1.2. Risk Factors

The research literature on teen sexting suggests that sexting behavior is likely to occur in various contexts and can be influenced by several factors [13]. Not surprisingly, exposure to pornography has reportedly increased the likelihood of engaging in sexting behavior, as indicated in several studies. For instance, a large survey of 4564 young people (ages 14–17) in five European countries reported that boys who regularly watched online pornography were more likely to send sexual images/messages [14]. In another study with a sample of 329 adolescents [15], pornography consumption was positively associated with sexting behavior for both boys and girls. 

Dating partners represent another potential risk factor that increases adolescents’ odds of engaging in sexting behavior, as studies show that an increasing number of young people increasingly develop and establish romantic relationships online [16,17,18]. According to Bianchi et al. [19], sexting between dating partners can be motivated by adolescents’ developmental needs, which include sexual expression (flirting), socialization needs, and self-expression, and online approval from peers or a dating partner. However, harmful motivations for sexting can be driven by coercive sexting among dating partners [20,21], as well as revenge toward an ex-partner [22].

Research exploring how bullying relates to sexting among teenagers is surprisingly limited. However, existing studies have consistently documented that adolescents who were involved in bullying (as a victim or perpetrator) were likely at risk of displaying sexting behavior [23,24,25]. For instance, a study conducted by Ojeda et al. [24] found that perpetrators of bullying (traditional and cyber) were at an increased risk of sexting (sending, receiving, third-party forwarding, and receiving sexts through an intermediary). However, only traditional bullying was associated with third-party forwarding of sexts without consent. Moreover, Ouytsel et al. [25] and Gamez-Guadix et al.’s [23] longitudinal studies also showed a reciprocal association between bullying victimization (traditional and cyberbullying) and sexting behavior (sending, receiving, or requesting sexual images). These findings suggest that involvement in bullying is a significant predictor of sexting behavior, which warrants closer research scrutiny.

Research has shown that delinquent and deviant behaviors [26,27], such as substance use [26,28], for example, are likely to elevate sexting risk in adolescents. To illustrate, Woodward et al.’s [26] findings indicated that delinquency, alcohol, marijuana use, and bullying increased adolescents’ odds of sending and receiving nude photos. The study also found that females who had participated in sexting behavior were more likely to consume alcohol while males were more likely to report using marijuana and being identified as bullies [26]. The association between delinquent and deviant behaviors and sexting can be supported by Gottfredson and Hirschi’s [29] proposition that low self-control, characterized by impulsivity, lack of diligence, risk-seeking, desire for physical activity, self-centeredness, and a low tolerance for frustration, motivates adolescents involved in delinquent behavior to seek deviant behavior such as sexting.

Consistent with literature indicating that gender and sexual minority youth are more likely to engage in risky sexual behaviors [30], this population has been found to be more likely to engage in sexting behaviors (sending or receiving sexually explicit pictures) as well [3]. The higher prevalence of sexting in this population may be explained by the fact that gender and sexual minority youth may feel as though they cannot be intimate with their partners in public, thus sexting could serve as a mean by which these youth can feel intimately connected and involved with their partners [4]. Sexual minority identification as a risk factor for engaging in sexting should be carefully interpreted to understand that it is not one’s sexual or gender minority identity that inherently positions them to be more likely to engage in this behavior, but the heteronormative context in which they exist, whereby systemic inequities hinder gender and sexual minority youth from a normative path of development regarding romantic and sexual relationships [31]. As such, developing inclusive practices and policies in schools may help to narrow the gap between lesbian, gay, bisexual, transgender, and questioning (LGBTQ) and heterosexual youth engaging in sexting behaviors.

### 1.3. Protective Factors

The research literature on protective factors that inhibit adolescent sexting behavior is limited. However, protective factors within various contexts have been widely considered in research on adolescent development and behaviors. Within the level of the family, there has been a long history of empirical studies that have identified the importance of parental monitoring, which is characterized by how parents oversee and regulate their adolescent children’s behaviors and activities [32]. However, it is unclear what parents can do to prevent their adolescent children’s engagement in norm-breaking and delinquent behaviors [33]. That said, research findings highlight the significant role that parental monitoring plays in adolescent behaviors, as studies have documented that youth whose parents monitor their activities consistently are less likely to engage in risky behaviors [34,35].

Social support is among the most widely recognized protective buffers in scholarship on adolescent behavior, as it plays a vital role in lowering youths’ odds of risky behaviors, such as bullying, substance use, and delinquent behaviors, as well as promoting mental health [36]. Social support refers to various kinds of supportive relationships or interactions that can promote an individual’s well-being by buffering against negative outcomes [37]. Family social support, in particular, is among the most widely researched protective factors as demonstrated in numerous empirical findings [38,39]. As Kashani et al. [36] purported, it is essential for children to have a sense that they are supported by their family to avoid behavioral problems [40]. A family that provides a high level of social support for adolescents can create a buffer against stressors [40,41], which is also likely to lower the likelihood of developing behavioral problems; thus, it is plausible that family social support is a vital protective buffer against sexting behavior in adolescents.

Although family support is of critical importance for teenagers, adolescence is a developmental period in which social support from friends and other adult figures is increasingly important. Similar to family support, social support from friends has been shown to have a positive impact on adolescent behaviors as it protects them from engaging in risky behaviors, substance use, bullying, and dating violence [42,43,44]. Moreover, support from adult figures (e.g., teachers) especially in school is another protective mechanism by which problematic behaviors can be avoided. According to one study, disruptive behavior was more likely to be exhibited by students in classrooms with low teacher support, as opposed to those in classrooms with high teacher support [45]. Hence, it is conceivable that adolescents who perceive their friends and teachers as showing support are less inclined to engage in misbehaviors including sexting.

Perceived school belonging is a protective buffer within the school context that has been widely researched in scholarship on adolescent behaviors. School belonging encompasses many connections and relationships that students establish within the school (e.g., extracurricular activities, relations with peers, etc.) [46]. School bonding not only promotes academic motivation and better physical health but also could play a critical role in inhibiting adolescents’ motivation to participate in risky behaviors, as research findings have shown [46,47]. Thus, school belonging might possibly inhibit adolescents’ propensity to participate in sexting behaviors.

### 1.4. The Current Study

The purpose of the present study is to identify and explore protective factors by which non-consensual sexting perpetration can be impeded by incorporating multiple domains of the social ecology and examining separately sexting perpetration among LGB and heterosexual students. The following protective factors were hypothesized to be associated with lower non-consensual sexting perpetration: empathy (individual level), family social support (family level), school belonging (school level), peer social support (school level), and adult social support (school level). Additionally, we hypothesize that the following risk factors would be associated with higher sexting: impulsivity (individual level), pornography exposure, number of dating partners, sexual activity, risky sexual behavior, alcohol and substance use, homophobic name calling perpetration, bullying perpetration, and delinquency.

## 2. Materials and Methods

### 2.1. Participants

This study was conducted with *N* = 2501 (LGB: *n* = 309, 76.4% female; non-LGB: *n* = 2192, 47.4% female) students from three Midwestern high schools. The schools were selected because the school district administration expressed interest in participating in the study. All students from each school were recruited. These public schools are situated in a Midwestern school district where 60.4% of the students are African American, followed by 31.5% European American, 2.6% Asian, 5.1% Hispanic, and 0.4% multi-racial. Approximately 69.3% of the student population is considered low-income. The chronic truancy rate for the school district is 2.5% and the mobility rate is 30.1%. The overall sample was 50.4% female, 49.1% African American, 34.3% white, 5.7% Hispanic, and 2.4% Asian/Pacific Islander and 8.4% other race. The mean age for the sample was 15.81 years (SD = 1.04) for non-LGB and 15.75 years (SD = 0.98) for LGB students, ages ranged from 11 to 19 years old. Regarding sexual attraction the sample was 86.3% exclusively hetero, 4.2% predominantly hetero, 2.0% bisexual but not hetero, 2.1% bisexual, 0.5% bisexual but more gay or lesbian, 0.3% predominantly lesbian or gay, and 0.6% exclusively lesbian or gay.

IRB Protocol Number: 08226.

### 2.2. Procedures

The university institutional review board and school district administration approved a waiver of active parental consent. Students were asked to assent to participate in the study through an assent procedure included on the cover sheet of the survey. A 95% participation rate was achieved. Six trained research assistants and the primary investigator collected data. Information about the authors and their relation to the study are provided in Appendix A.

### 2.3. Measures

Students completed a self-report survey on computers in school computer labs. The survey consisted of demographics as well as measures assessing sexting perpetration and risk and protective factors associated with these behaviors across several social ecological domains.

### 2.4. Demographic Variables

Demographic information was collected, including gender, age, sexual orientation, and race. For the race, participants were given five options: African American (not Hispanic), Asian, white (not Hispanic), Hispanic, and other (with a space to write in the preferred descriptor). Sexual orientation was measured with seven options: exclusively hetero, predominantly hetero, bisexual but not hetero, bisexual, bisexual but more gay or lesbian, predominantly lesbian or gay, and exclusively lesbian or gay. The sexual orientation scale was adapted from the Kinsey scale of sexual orientation which is designed to measure a continuum from heterosexual to exclusively gay/lesbian sexual orientation [48,49]. The options for sexual orientation were recoded as LGB for any option except “exclusively hetero”, which was coded as Heterosexual.

### 2.5. Dependent Variable

*Sexting perpetration.* Cyber sexting was assessed with a four-item scale based on the work of Ybarra, Espelage, and Mitchell [50] and one item from the American Association of University Women (AAUW) Sexual Harassment Survey—Perpetration Scale [51,52]. Students were asked how often they did the following things in the last school year: “tried to get someone else to talk about sex online when they did not want to”; “asked someone to do something sexual online when the other person did not want to”; “sent a picture text message that was sexual in any way when that person did not want to receive it”; and the AAUW item was “showed them sexy or sexual pictures that they didn’t want to see”. Response options included “Never”, “Rarely”, “Occasionally”, and “Often” on a Likert-type scale. Items were averaged and higher scores indicate greater sexting. Cronbach’s alpha coefficient for the current study was 0.80.

### 2.6. Independent Variables

#### Individual Domain

*Empathy.* This 5-item scale of the Teen Conflict Scale [53,54] measures adolescents’ ability to listen, care, and trust others. Students are asked to indicate how often they would use items in the scale to describe themselves. Examples include: “I can listen to others”, and “I get upset when my friends are sad”. Response options are recorded on a 5-point Likert-type scale and are “Never”, “Seldom”, “Sometimes”, “Often”, and “Always”. Items were averaged and higher scores indicate greater empathy. In the current study, we found the scale to have a Cronbach’s alpha coefficient 0.76.

*Impulsivity.* The 4-item Impulsivity subscale from the Teen Conflict Survey [53] assesses the self-reported impulsivity of the respondents. Students are asked how often they would say the following statements about themselves: “I have a hard time sitting still”, “I start things but have a hard time finishing them”, “I do things without thinking”, and “I need to use a lot of self-control to keep out of trouble”. Response options include “Never”, “Seldom”, “Sometimes”, “Often”, and “Always” on a Likert-type scale. Items were averaged and higher scores indicate greater impulsivity. A Cronbach alpha of 0.62 to 0.76 was recorded in previous studies [53,55]. In the current study, a Cronbach’s alpha coefficient of 0.76 was found.

### 2.7. Family Domain

*Family Social Support.* Family social support was measured using the 3-item family subscale from the Vaux Social Support Record. The VSSR is a 9-item questionnaire that is an adaptation of Vaux’s [56] Social Support Appraisals (SSA) 23-item scale that was designed to assess the degree to which a person feels cared for, respected, and involved [56]. The family subscale is three items that measure the support available from the family. A sample item is “There are people in my family I can talk to, who care about my feelings and what happens to me”. The family subscale showed good internal consistency across samples. Responses are “Not at all”, “Some” or “A lot” on a Likert-type scale. Items were averaged and higher scores indicate greater social support. Cronbach’s alpha coefficient 0.90 was found for this study.

*Parental Monitoring*. The Parental Supervision subscale from the Seattle Social Development Project [57] was used to measure perceptions of established familial rules and perceived parental awareness regarding schoolwork and attendance, peer relationships, alcohol or drug use, and weapon possession. The subscale includes 8 items and response options include “Never”, “Seldom”, “Sometimes”, “Often”, and “Always” on a Likert-type scale. Items were averaged and higher scores indicate greater parental monitoring. Example items include, “My family has clear rules about alcohol and drug use” and, “My parents ask if I’ve gotten my homework done”. In the current study, we found the scale to have a Cronbach’s alpha coefficient of 0.86.

### 2.8. School Domain

*School belonging.* A shortened 4-item version of the Psychological Sense of School Membership (PSSM [58]) was used to assess students’ sense of belonging or psychological membership in his or her school (i.e., the extent to which middle school students feel personally accepted, respected, included, and supported by others in the school; example items include “Other students in this school take my opinions seriously”, “The teachers here respect me”). Participants responded to the items using a 4-point Likert-type scale, including “Strongly disagree”, “Disagree”, “Agree”, and “Strongly agree” and higher scores reflect a stronger sense of school belonging. A Cronbach’s alpha coefficient 0.64 was found for this study.

*Adult social support.* Adult social support was measured using the 3-item adult social support subscale from the Vaux Social Support Record. The VSSR is a 9-item questionnaire that is an adaptation of Vaux’s [56] Social Support Appraisals (SSA) 23-item scale that was designed to assess the degree to which a person feels cared for, respected, and involved [56]. The adult subscale is three items that measure the support available from the adults at school. A sample item is “at school, there are adults I can talk to, who care about my feelings and what happens to me”. The adult subscale showed good internal consistency across samples. Responses are “Not at all”, “Some” or “A lot” on a Likert-type scale. Items were averaged and higher scores indicate greater adult social support. Cronbach’s alpha coefficient 0.87 was found for this study.

*Peer social support.* Friend social support was measured using the 3-item family subscale from the Vaux Social Support Record. The VSSR is a 9-item questionnaire that is an adaptation of Vaux’s [56] Social Support Appraisals (SSA) 23-item scale that was designed to assess the degree to which a person feels cared for, respected, and involved [56]. The friend subscale is three items that measure the support available from friends. A sample item is “I have friends I can talk to, who give good suggestions and advice about my problems”. The friend subscale showed good internal consistency across samples. Responses are “Not at all”, “Some” or “A lot” on a Likert-type scale. Items were averaged and higher scores indicate greater peer social support. Cronbach’s alpha coefficient 0.92 was found for this study.

### 2.9. Risky Behaviors

*Pornography exposure.* Two items assessed pornography exposure: (1) Have you visited a sexually explicit website during the past 30 days? (2) Have you ever read a pornographic magazine, seen a pornographic film, or pornography on the internet? Response options were “No” or “Yes”. Items were averaged and higher scores indicate greater social pornography exposure. These two items were highly correlated (r = 0.84).

*Number of dating partners.* One item was used to assess the number of dating partners: How many people have you dated seriously? Response options ranged from 0 (None) to 10 (10 or more).

*Sexual activity.* Sexual activity was measured with three items that asked the frequency of (1) having oral sex, (2) having vaginal intercourse, and (3) having anal sex. Response options were “Never in my life”, “Not in the past 12 months”, “Seldom”, “Sometimes”, “Frequently”, and “Very frequently” on a Likert-type scale. Items were averaged and higher scores indicate greater sexual activity. Cronbach alpha coefficient was 0.61 for the current study.

*Risky Sexual Behavior.* Two items were used to measure risky sexual behaviors: (1) drank alcohol before or during sex, and (2) used marijuana or drugs before or during sex. Response options were “Never in my life”, “Not in the past 12 months”, “Seldom”, “Sometimes”, “Frequently”, and “Very frequently” on a Likert-type scale. Items were averaged and higher scores indicate greater risky sexual behavior. Cronbach alpha was 0.75 for the current study.

*Alcohol and drug use.* An eight-item scale asked students to report how many times in the past 30 days they used alcohol or drugs in Wave 1 [59]. The scale included statements like “drunk beer”, “smoked cigarettes”, “drunk liquor”, and “used marijuana”. Response options include “0 days, “1 day”, “2 days”, “3–5 days”, “6–9 days”, “10–19 days”, and “20–30 days” on a Likert-type scale. Items were averaged and higher scores indicate greater alcohol and drug use. The scale correlates positively with risk behaviors like delinquency and correlates negatively with positive behaviors including school attendance [59]. Farrell et al. [59] reported a Cronbach’s alpha of 0.87 with a sample of urban adolescents. The final two items (“used inhalants” and “used other drugs”) were not used in the analysis due to very low endorsement of these behaviors, which was not surprising given that these items were not as developmentally appropriate for this age group. Cronbach’s alpha coefficient for the current study was 0.74.

### 2.10. Violence Perpetration

*Homophobic bullying perpetration.* The 5-item Homophobic Content Agent Scale was used to assess homophobic teasing perpetration [60]. Students were asked how often in the past 30 days they directed homophobic epithets at other students. Students were asked, “how many times in the last 30 days did YOU say [homo, gay, lesbo, or fag] to” various categories of peers for each item. Response options included “Never”, “1 or 2 times”, “3 or 4 times”, 5 or 6 times”, and “7 or more times” on a Likert-type scale. Items were averaged and higher scores indicate greater homophobic bullying. Construct validity has been supported through exploratory and confirmatory analyses [60]. Cronbach’s alpha coefficient was 0.79 for this study.

*Bullying perpetration.* The nine-item Illinois Bully Scale [61] assesses the frequency of bullying at school. Students are asked how often in the past 30 days they did the following to other students at school: teased other students, upset other students for the fun of it, excluded others from their group of friends, helped harass other students, and threatened to hit or hurt another student. Response options include “Never”, “1 or 2 times”, “3 or 4 times”, “5 or 6 times”, and “7 or more times” on a 5-point Likert-type scale. Items were averaged and higher scores indicate greater bullying. The construct validity of this scale has been supported via exploratory and confirmatory factor analysis [61]. The scale correlated moderately with the Youth Self-Report Aggression Scale (r = 0.65; [62]), suggesting that it was somewhat unique from general aggression. Concurrent validity of this scale was established with significant correlations with peer nominations of bullying [63]. Cronbach’s alpha coefficient was 0.84 for this study.

*Delinquency.* A 7-item scale is based on Jessor and Jessor’s [64] General Deviant Behavior Scale and asks students to report how many behaviors listed on the measure they took part in during the last year. The scale consists of items such as, “Skipped school”, and “Damaged school or other property that did not belong to you”. Response options include “Never”, “1 or 2 times”, “3 or 5 times”, “6 or 9 times”, and “10 or more times” on a Likert-type scale. Items were averaged and higher scores indicate greater delinquency. The original study by Jessor and Jessor [64] utilized this scale in a longitudinal study of 432 largely white middle-class students in 7th–10th grades. A mean Cronbach’s alpha coefficient of 0.76 was reported across the 3-year study. Since its development, this scale has been used numerous times resulting in Cronbach’s alpha coefficients ranging from 0.76 to 0.83 [59]. A Cronbach’s alpha coefficient 0.67 was found for this study.

### 2.11. Analysis Plan

We conducted a series of path analysis models for each domain of the social ecology and risky behaviors. Each model was run separately for LGB students and non-LGB (heterosexual) students. Model 1 included only the demographic indicators age, female, and African American. All subsequent models also controlled for demographic indicators while adding distinct predictors for each domain. Model 2 added the individual-level predictors empathy and impulsivity. In Model 3, the family-level predictors family social support and parental monitoring were added. Similarly, Model 4 added the school-level predictors school belonging, peer social support, and adults-at-school social support. Next, Model 5 included the risky behaviors pornography exposure, number of dating partners, sexual activity, risky sexual behaviors, and alcohol and substance use. Lastly, Model 6 included the predictors homophobic bullying perpetration, bullying perpetration, and delinquency. Because all variables included in the path analysis models were observed scale indicators (not latent), each model was just identified and had a perfect fit to the data (CFI = 1, TLI = 1, RMSEA = 0).

Additionally, we conducted a moderation analysis for both groups with the set of protective factors and risk factors that emerged as significantly associated with sexting perpetration in the first set of models. For the interaction analysis, the final model was created by selecting only the significant predictors across all models forming a more parsimonious combined model (see Figure 1). Predictors were centered and interaction terms were created by multiplying each predictor (risk factor) and moderator (protective factor) before entering them into the final model. Lastly, significant interaction terms were plotted to facilitate the interpretation of the interactions.

All analyses were run with the statistical software R and the structural equation modeling package Lavaan [65]. Full Information Maximum Likelihood (FIML) was employed to account for missing data [66]. Finally, descriptive statistics and bivariate correlations were computed for each outcome.

## 3. Results

Table 1 presents descriptive statistics for each outcome disaggregated by LGB and non-LGB groups. Appendix A presents bivariate correlations for each outcome. Descriptive statistics suggest that LGB students had only slightly higher levels of sexting perpetration when compared to heterosexual students. Similarly, LGB students had higher levels of risky behaviors, such as pornography exposure, number of dating partners, sexual activity, bullying, delinquency, and alcohol and substance use when compared to heterosexual students. Regarding protective factors, LGB students had slightly lower levels of school belonging, peer social support, adult social support, family social support, and parental monitoring when compared to heterosexual students.

Model 1 added student demographics to the regression of sexting perpetration (Table 2). Results of Model 1 showed a significant association for female non-LGB (b = −0.03, S.E. = 0.01, *p* < 0.05), and African American non-LGB students (b = 0.03, S.E. = 0.01, *p* < 0.05). That is, among non-LGB students, being female was associated with lower sexting perpetration, and being African American was associated with higher sexting perpetration. None of the demographic variables were statistically significant among LGB students.

Model 2 added the individual level predictors empathy and impulsivity (Table 2). Neither empathy nor impulsivity was significantly associated with sexting perpetration among LGB or non-LGB students.

Model 3 added the family-level predictors family social support and parental monitoring (Table 2). A significant negative association between parental monitoring and sexting perpetration (b = −0.08, S.E. = 0.03, *p* < 0.01) was detected among LGB students. These findings suggest that higher parental monitoring among LGB students is associated with lower sexting perpetration.

Model 4 added the school level predictors school belonging, peer social support, and adult social support (Table 2). None of the school-level predictors were significantly associated with sexting perpetration for LGB or non-LGB students.

Model 5 added the risky behaviors pornographic exposure, number of dating partners, sexual activity, risky sexual behaviors, and alcohol and substance use (Table 2). A statistically significant association was found between higher pornography exposure and sexting perpetration among LGB students (b = 0.13, S.E. = 0.05, *p* < 0.01). Similarly, a higher number of dating partners was statistically significantly associated with higher sexting perpetration among LGB students (b = 0.01, S.E. = 0.01, *p* < 0.05). Lastly, higher alcohol and substance use was statistically significantly associated with higher sexting perpetration among non-LGB students (b = 0.05, S.E. = 0.01, *p* < 0.001).

Model 6 added the predictors homophobic bullying perpetration, bullying perpetration, and delinquency (Table 2). Positive and statistically significant associations were found for both groups in the predictors bullying perpetration (LGB: b = 0.17, S.E. = 0.04, *p* < 0.001; non-LGB: b = 0.08, S.E. = 0.01, *p* < 0.001) and delinquency (LGB: b = 0.13, S.E. = 0.04, *p* < 0.001; non-LGB: b = 0.06, S.E. = 0.02, *p* < 0.001). That is, for both groups, higher bullying and delinquency were associated with higher sexting perpetration.

Table 3 presents the moderation analysis between parental monitoring as a moderator and the significant risk factors of bullying perpetration, delinquency, pornography exposure, number of dating partners, and alcohol and substance use. The same set of predictors were tested for both groups. Results suggested that parental monitoring significantly moderated the association between bullying perpetration and sexting perpetration for LGB students (b = −0.07, S.E. = 0.01, *p* < 0.001). Additionally, parental monitoring significantly moderated the association between the number of dating partners and sexting perpetration among LGB students (b = −0.05, S.E. = 0.01, *p* < 0.001). Lastly, parental monitoring significantly moderated the association between alcohol and substance use and sexting perpetration for both groups (LGB: b = 0.05, S.E. = 0.01, *p* < 0.001; non-LGB: b = −0.02, S.E. = 0.01, *p* < 0.01).

Figure 2 presents the plotted interaction between bullying perpetration and parental monitoring with the outcome of sexting perpetration among LGB students. The plot suggests that, among LGB students with higher bullying perpetration (+1 SD), those with lower parental monitoring (−1 SD) are associated with higher sexting when compared to LGB students with higher parental monitoring (+1 SD). However, the opposite seems to occur among LGB students with low bullying perpetration (−1 SD), where those with higher parental monitoring (+1 SD) are associated with higher sexting when compared to students with lower parental monitoring (−1 SD).

A similar pattern emerged in Figure 3 for the moderation of the number of dating partners one reported and parental monitoring for LGB students. Among LGB students with a higher number of dating partners (+1 SD), those with low parental monitoring (−1 SD) were associated with higher sexting perpetration when compared with students with higher parental monitoring (+1 SD). Conversely, among LGB students with a low number of dating partners (−1 SD), those with high parental monitoring (+1 SD) were associated with higher sexting perpetration when compared with students with low parental monitoring (−1 SD).

Figure 4 presents the interaction between alcohol and substance use and parental monitoring for LGB students while Figure 4 presents the same interaction for non-LGB students. Figure 3 suggests that among LGB students with high alcohol and substance use (+1 SD), those with higher parental monitoring (+1 SD) are associated with higher sexting perpetration when compared to those with lower parental monitoring (−1 SD). In contrast, among LGB students with lower alcohol and drug use (−1 SD), students with higher parental monitoring (+1 SD) are associated with lower sexting perpetration when compared to students with low parental monitoring (−1 SD).

Lastly, this pattern was reversed among non-LGB students (Figure 5). For non-LGB students with higher alcohol and substance use (+1 SD), those with higher parental monitoring (+1 SD) are associated with slightly lower sexting perpetration when compared to those with low parental monitoring (−1 SD). Conversely, among non-LGB students with low alcohol and drug use (−1 SD), students with high parental monitoring (+1 SD) are associated with higher sexting perpetration when compared with students with lower parental monitoring (−1 SD).

## 4. Discussion

The present study tested the associations between risk and protective factors for sexting perpetration among groups of LGB and heterosexual students across multiple domains of youth’s social ecology. A significant association was found between higher parental monitoring and lower sexting perpetration among LGB students. This finding expands our understanding of parental monitoring among a sample of sexual minority youth and is consistent with previous findings showing a negative association between parental monitoring and risky sexual behaviors [67,68,69,70]. Our study suggests that positive parental monitoring and involvement in youths’ activities could have a protective effect on sexting perpetration, particularly among LGB adolescents. The effect of parental monitoring could be due to higher supervision of cellphone and internet use among more involved parents which could directly be associated with youths’ use of cellphones for sexting [71]. Additionally, this finding could be explained by higher adolescent-parent communication and a better family dynamic among LGB students with higher parental monitoring which would support the establishment of clear rules around sexting and other risky behaviors [70]. Given that LGB youth have been associated with higher use of social media and technology for dating and to explore their sexuality when compared to non-LGB adolescents, the role of parental monitoring among LGB students is of utmost importance for the prevention of sexting perpetration [72,73].

Additionally, moderation analyses among LGB students revealed that parental monitoring moderated the associations between sexting and the risk factors of bullying perpetration and number of dating partners. Among students who showed higher engagement in bullying and higher number of dating partners, higher parental monitoring was associated with lower sexting perpetration. However, at lower levels of the risk factors, higher parental monitoring was associated with higher sexting. These findings suggest that higher parental monitoring may have the greatest benefits among LGB students who already show higher incidence of risky behaviors and could potentially be harmful among non-involved students as some literature suggests that too much parental monitoring can have adverse effects on parents’ relationship with their adolescents [74].

Although parental monitoring was not directly associated with sexting among heterosexual students, we found evidence of moderation effects between alcohol and drug use and parental monitoring among this population. These findings contextualize the protective effect of parental monitoring among heterosexual students, showing that among those with high use of drugs and alcohol, having higher parental monitoring was associated with lower sexting perpetration. However, the opposite was true among heterosexual students reporting lower alcohol and substance use. It may be that excessive parental monitoring could also have a detrimental effect among heterosexual students who do not show elevated risk factors such as alcohol and substance abuse but would be more appropriate when students show higher involvement in such behaviors [74].

However, when examining the moderation between parental monitoring and alcohol and substance use among LGB students, the opposite pattern emerged. Students with lower levels of substance use but higher parental monitoring had lower sexting perpetration when compared to students with similar levels of substance use and lower parental monitoring. These results suggest that the role of parental monitoring may vary not only according to the levels of involvement of youth in risky behaviors, but also the type of risky behavior and the sexual orientation of the participant. Therefore, we recommend that future studies delve deeper into the role of parental monitoring in moderating the risk factors of sexting perpetration and consider other markers of sexual and gender identity in their analysis.

Significant associations were also found between the risk factors pornography exposure and number of dating partners among LGB students. These results support previous evidence showing an association between pornography exposure and sexual risk taking [75,76], such as early initiation of sexual activity [77,78] and sexting [79,80]. We expand on these findings by providing evidence of support for this association among LGB students as well. It has been proposed that pornography exposure could be associated with youth trying to re-enact sexually explicit material, and therefore be associated with higher sexting perpetration [80], thus helping to contextualize this finding. Regarding number of dating partners, LGB youth may be more likely to use sexting to communicate with their own romantic partners when compared to heterosexual youth because it may be harder for them to find a romantic partner nearby [81]. Additionally, this association may be due to LGB students who date more often having a higher degree of sexual sensation seeking in general when compared to non-dating LGB students [76]. In light of these findings, more research should be dedicated to understanding the sexual development of sexual minority students and how to prevent sexual risk-taking behaviors among this population.

Additionally, the risk factor alcohol and substance use was significantly associated with sexting among heterosexual students. This finding replicates previous studies showing a relationship between alcohol use and other risky sexual behaviors such as sexting [80,82,83]. Alcohol use, in particular, has been associated with higher sexual disinhibition, sexual risk taking, and subjective arousal which could explain the association between alcohol use and sexting perpetration [83]. This finding suggests that prevention programs that target substance use behaviors could have an indirect effect on sexual risk taking and sexting. These insights could be leveraged in universal prevention programs which could perhaps include sexual risk-taking prevention in their curriculum given the consistent evidence for the association between these constructs.

Finally, the risk factors of bullying perpetration and delinquency were also significantly associated with higher sexting among both heterosexual and LGB students. These findings are consistent with the literature reviewed showing that bullying and delinquency may play a role in involvement in sexting [23,24,25,26]. These findings could be explained through the bullying to sexual violence pathway wherein youth who bully their peers or engage in delinquent behaviors may be more likely to also engage in sexual violence or sexual risk taking like sexting to assert dominance and power over their peers [51].

### Limitations

The present study is an important contribution to the sexting literature due to the large sample of LGB students and a comprehensive examination of risks and protective factors at multiple levels of youth social ecology. However, these results should be interpreted cautiously due to noteworthy limitations. First, the cross-sectional nature of the data does not allow us to determine the directionality of the associations uncovered in this paper. Future studies should employ a longitudinal design when examining the significant risks and protective factors explored in this paper. Second, an important limitation is that the sexting variable used in this study did not address whether sexting occurred because of malicious intent (e.g., revenge sexting, sending sexts to a stranger) or among dating partners, which confounds both types of sexting. Third, the LGB and non-LGB groups were not equivalent in size given the sample was recruited from general school-based populations. Future research would need to oversample LGB youth to address this limitation. Finally, there are other potential predictors of sexting involvement that were not examined that future research should consider (e.g., family dynamics, mental health symptoms).

## 5. Conclusions

The present study provides noteworthy directions for practice and future research. First, it is important to acknowledge that sexual exploration and development is a normative process throughout the adolescent years [17], and these findings should not be interpreted to cast inherently negative judgement on the sexual behaviors of adolescents. However, the associations from the current study between sexting and other risky behaviors are consistent with extant literature and emphasize the need to intervene and prevent the occurrence of non-consensual sexting perpetration [23,24,25,26]. To prevent sexting and other risky behaviors practitioners should consider the protective factors of sexting perpetration such as encouraging appropriate levels of parental monitoring, in addition to the implementation of bullying and alcohol and drug prevention programming. As youth begin to explore their sexuality it is important that supportive adults and practitioners are prepared to engage in discussions around healthy relationships, consent, and sex. This is especially important among sexual and gender minority youth who receive little to no guidance on navigating non-heteronormative relationships. Furthermore, involvement in sexting can have serious legal consequences for youth that could change the entire trajectories of their lives. As such, it is imperative that youth understand the consequences of sexting behaviors and the laws regarding the possession and distribution of nude photos displaying a minor (i.e., child pornography). The findings in the current study indicate the importance of capturing unique individual and contextual factors within the social ecology that influence perpetration of non-consensual sexting for LGB and non-LBG youth.

Future research should consider the developmental trajectories of adolescent sexuality as it relates to teen sexting, especially among sexual minority youth where the literature is less conclusive. Given that sexual minority youth may experience more difficulty finding a romantic partner, future studies should emphasize how this may play a role in their online behavior [30,31]; it is possible their trajectory of sexual development throughout adolescence may be different than their heterosexual peers in regard to online behavior in particular. Additionally, parental monitoring continues to be an ambivalent concept in the literature on youth risk-taking and prevention, such that it seems to be both protective and inhibitive depending on the demographics of the youth and the degree of parental monitoring [32,33,34,35]. Future research should continue to investigate this concept so that more conclusive recommendations can be made.

## Figures and Tables

**Figure 1 ijerph-17-09477-f001:**
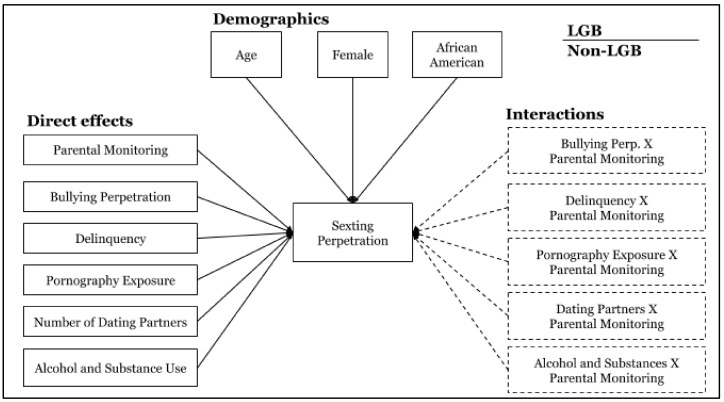
Final path analysis model.

**Figure 2 ijerph-17-09477-f002:**
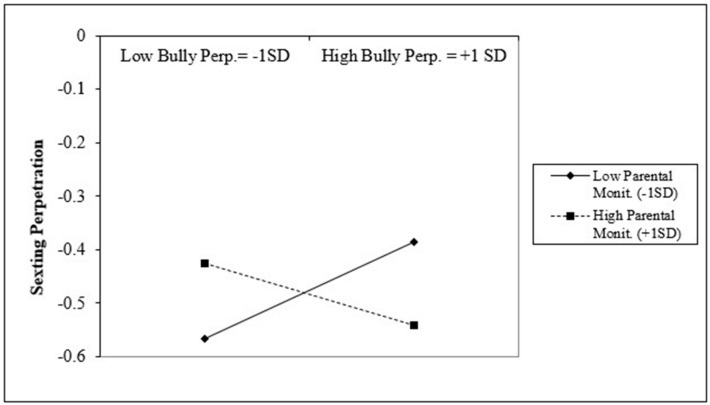
Interaction between bullying perpetration and parental monitoring for LGB group.

**Figure 3 ijerph-17-09477-f003:**
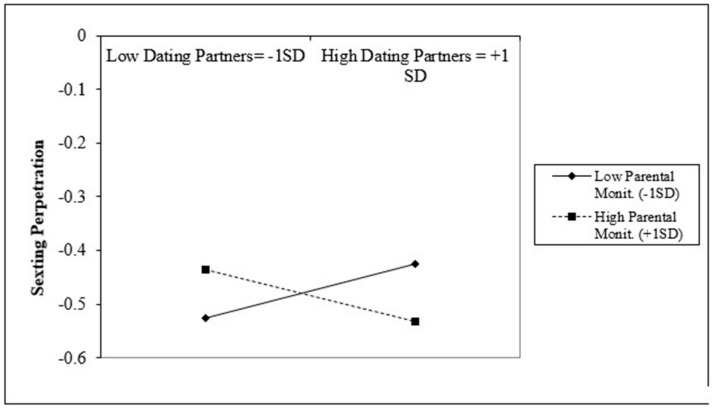
Interaction between number of dating partners and parental monitoring for LGB group.

**Figure 4 ijerph-17-09477-f004:**
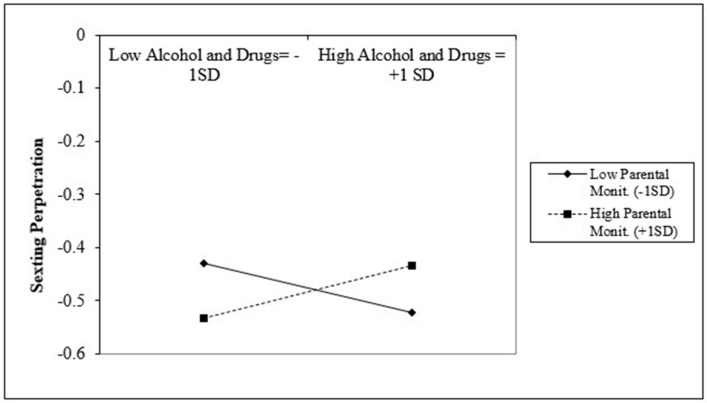
Interaction between alcohol and drug use and parental monitoring for LGB group.

**Figure 5 ijerph-17-09477-f005:**
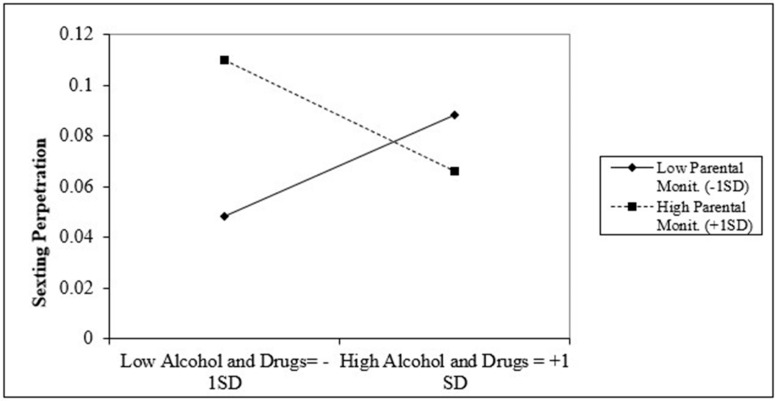
Interaction between alcohol and drug use and parental monitoring for LGB group.

**Table 1 ijerph-17-09477-t001:** Descriptive statistics.

Variable	LGB (*n* = 309)		Non-LGB (*n* = 2192)		
	M/*n*	SD/%	M/*n*	SD/%	Range
*Demographics*					
Age	15.75	0.98	15.81	1.04	11–19
Female	236	76.4%	1035	47.4%	
African American	129	48.3%	1018	50.1%	
White	93	34.8%	665	32.7%	
Other	45	16.9%	348	15.9%	
*Dependent variable*					
Sexting perpetration	0.04	0.26	0.03	0.23	0–3
*Individual level*					
Empathy	2.05	0.94	2.03	0.85	0–4
Impulsivity	1.41	0.93	1.46	0.95	0–4
*Family level*					
Family social support	1.20	0.66	1.45	0.60	0–2
Parental monitoring	2.05	0.75	2.20	0.70	0–3
*School level*					
School belonging	1.86	0.57	2.03	0.53	0–3
Peer social support	1.35	0.58	1.39	0.58	0–2
Adult social support	0.99	0.59	1.04	0.57	0–2
*Risky behaviors*					
Pornography exposure	0.25	0.40	0.22	0.38	0–1
Number of dating partners	3.26	2.88	2.93	2.84	0–10
Sexual activity	1.42	1.05	1.37	1.05	0–5
Risky sexual behaviors	0.45	0.94	0.50	0.96	0–5
Alcohol and substance use	0.66	0.97	0.32	0.69	0–6
*Violence perpetration*					
Homophobic bullying perp.	0.56	0.80	0.57	0.76	0–4
Bullying perpetration	0.49	0.59	0.40	0.53	0–4
Delinquency	0.51	0.61	0.35	0.44	0–4

**Table 2 ijerph-17-09477-t002:** Results of structural equation models, outcome sexting perpetration.^1^

Variable	LGB (*n* = 309)	Non-LGB (*n* = 2192)
	b (S.E.)	b (S.E.)
*Demographics (Model 1)*		
Age	0.01 (0.02)	−0.00 (0.01)
Female	−0.07 (0.05)	**−0.03 (0.01)** *
African American	0.01 (0.04)	**0.03 (0.01)** *
*Individual level (Model 2)*		
Empathy	0.02 (0.02)	−0.01 (0.01)
Impulsivity	0.02 (0.02)	−0.00 (0.01)
*Family level (Model 3)*		
Family social support	−0.04 (0.05)	−0.01 (0.01)
Parental monitoring	**−0.08 (0.03)** **	−0.01 (0.01)
*School level (Model 4)*		
School belonging	−0.00 (0.04)	−0.01 (0.01)
Peer social support	−0.04 (0.11)	−0.01 (0.01)
Adult social support	−0.06 (0.05)	0.00 (0.01)
*Risky behaviors (Model 5)*		
Pornography exposure	**0.13 (0.05)** **	0.02 (0.02)
Number of dating partners	**0.01 (0.01)** *	0.00 (0.002)
Sexual activity	−0.01 (0.02)	−0.01 (0.01)
Risky sexual behaviors	0.00 (0.02)	0.00 (0.01)
Alcohol and substance use	−0.01 (0.02)	**0.05 (0.01)** ***
*Violence perpetration (Model 6)*		
Homophobic bullying perp.	−0.01 (0.02)	0.01 (0.01)
Bullying perpetration	**0.17 (0.04)** ***	**0.08 (0.01)** ***
Delinquency	**0.13 (0.04)** ***	**0.06 (0.02)** ***

^1^ *** *p* < 0.001, ** *p* < 0.01, * *p* < 0.05; Bold are significant associations.

**Table 3 ijerph-17-09477-t003:** Moderation analysis between parental monitoring and bullying and parental monitoring and delinquency.^1^

Variable	LGB (*n* = 309)	Non-LGB (*n* = 2192)
	b (S.E.)	b (S.E.)
*Demographics*		
Age	**0.03 (0.01)** **	−0.00 (0.01)
Female	−0.04 (0.03)	−0.01 (0.01)
African American	−0.03 (0.03)	0.02 (0.01)
*Direct effects*		
Parental monitoring	−0.00 (0.01)	0.01 (0.01)
Bullying perp.	0.02 (0.02)	0.04 (0.01)
Delinquency	0.02 (0.02)	0.02 (0.01)
Pornography exposure	0.01 (0.01)	0.01 (0.01)
Number of dating partners	0.00 (0.01)	0.01 (0.01)
Alcohol and substance use	0.00 (0.01)	−0.00 (0.01)
*Interactions*		
Bullying perp. X parental monitoring	**−0.07 (0.01)** ***	−0.01 (0.01)
Delinquency X parental monitoring	−0.01 (0.01)	0.00 (0.01)
Pornography exposure X parental monitoring	−0.00 (0.02)	−0.01 (0.01)
Dating partners X parental monitoring	**−0.05 (0.01)** ***	0.00 (0.01)
Alcohol and substances X parental monitoring	**0.05 (0.01)** ***	**−0.02 (0.01)** **

^1^ *** *p* < 0.001, ** *p* < 0.01, * *p* < 0.05; Bold are significant associations.

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
