# Peer review of "Social-Ecological Examination of Non-Consensual Sexting Perpetration among U.S. Adolescents"

_ijerph, 2020, doi:10.3390/ijerph17249477_

Round 1

Reviewer 1 Report

The Authors explore individual and contextual risk and protective factors that are associated with sexting behavior among a sample of adolescents.

The literature review is wide and well described, as the methods and results. 

Statistics is clear and complete.

Limits should be improved: many other intrinsic and extrinsic factors may influence behavior, including psychiatric conditions, mental retardation, adjustment disorders (not mentioned), as well as other family conditions (not only economical, but also family foster conditions or separated parents and so on...

It should be specified that the LGB sample (that gave the most interesting findings) was not proportional to the non-LGB group and was quite small.

Line 35: please specify what Nation (U.S.)

Author Response

Please see attached changes

Reviewer 2 Report

Provide a summary statement of all the data instruments before going into them in detail.

Please add an about the researcher's section so you tell us more about you as the researcher(s) and your connection to this study. How does this align with personal interests, professional work, etc., to help the reader place you directly in the center of your work?

Consider adding definitions/descriptions in the Methods. 

The conclusion should be supported with the literature and referenced to findings.

Author Response

Please see attached changes

Reviewer 3 Report

This paper focused on sexting which is a risky sexual behaviour among adolescents. Sexting is associated with adverse psychosocial outcomes such as depression, anxiety and other health problems. This paper is well structured according to the journal’s guideline. There are some revisions that need to be addressed before reviewer could give a decision for this paper.

Major revision

  1. Please include the SEM graph for the final SEM model because it would give better explanation especially if authors used SEM analysis.
  2. Please give more detailed explanation about the sample selection procedures. How authors choose the three school? Is it using a random sampling technique? Do the authors collect the data from all students in the school? What is the formula used to calculate the minimum sample size?
  3. How did the authors choose the final model? What kind of goodness of fit test used by the authors for the SEM model?

Minor revision:

  1. Please explain in more detail about how authors delivered the questionnaire. Is it by mail? Completed in the classroom? Or other type of delivery.
  2. Please give relevant literature that authors used as reference for sexual identity.
  3. Please give detailed explanation about the scoring method for each questionnaire. Is it using a Likert score? Because in some instrument (for example in line 211-225), authors did not mention about the score for each answer.

Author Response

Please see attached changes

Round 2

Reviewer 3 Report

All my suggestions/comments have been addressed. Thank you.